# Bioaccessibility Evaluation of Soymilk Isoflavones with Biotransformation Processing

**DOI:** 10.3390/foods12183401

**Published:** 2023-09-12

**Authors:** Gabriela Alves Macedo, Cíntia Rabelo e Paiva Caria, Paula de Paula Menezes Barbosa, Marina Rodrigues Mazine, Alessandra Gambero

**Affiliations:** 1Department of Food Science and Nutrition, School of Food Engineering, State University of Campinas (UNICAMP), Campinas 13083-862, SP, Brazil; 2Life Science Center, Pontifical Catholic University of Campinas (PUCCAMP), Campinas 13060-904, SP, Brazil

**Keywords:** polyphenols, β-glucosidase, probiotics, gastrointestinal, digestion *in vitro*

## Abstract

Soy isoflavones are considered important sources of bioactive compounds, but they are poorly absorbable, due to their large hydrophilic structures. Some biotransformation strategies have been used to convert the glycosidic form into aglycones, making them available for absorption. This study evaluated the potential of enzymatic and/or microbial fermentation combined bioprocesses in a soymilk extract before and after gastrointestinal in vitro digestion. Commercial β-glucosidase (ET) and a mix of commercial probiotics (F) containing *Lactobacillus acidophilus*, *Lactobacillus casei*, *Lactococcus lactis*, *Bifidobacterium bifidum*, and *Bifidobacterium lactis* were used to biotransform the soymilk phenolic extract. An isoflavone profile was identified using HPLC-DAD, total phenolic content was identified using the Folin–Ciocalteu test, and antioxidant capacity was identified using ORAC and FRAP. Soymilk enzymatically treated (ET) followed by microbial fermentation (ET + T) resulted in better conversion of glycosylated isoflavones (6-fold lower than control for daidzin and 2-fold for genistin) to aglycones (18-fold greater than control for dadzein and genistein). The total phenolic content was increased (3.48 mg/mL for control and 4.48 mg/mL for ET + T) and the antioxidant capacity was improved with treatments of ET + T (120 mg/mL for control and 151 mg/mL with ORAC) and with FRAP (285 µL/mL for control and 317 µL/mL). After the *in vitro* digestion, ET + T samples resulted in a higher content of genistein (two-fold higher than control); also, increases in the total phenolic content (2.81 mg/mL for control and 4.03 mg/mL for ET + T) and antioxidant capacity measured with ORAC were greater compared to undigested samples. In addition, the isolated microbial fermentation process also resulted in positive effects, but the combination of both treatments presented a synergistic effect on soy-based products.

## 1. Introduction

Soy isoflavones are a class of phytoestrogens that provide health benefits for humans [1]. However, the isoflavones present in soy products are usually found in the glycosylated form (daidzin, genistin, and glycitin), presenting low availability that is poorly absorbable by the gastrointestinal tract. For isoflavones to exert their health effects, the glycoside forms may be deconjugated by β-glucosidases produced by the intestinal microbiota, forming aglycone isoflavones (daidzein, genistein, and glycitein) and their secondary metabolites, which can be absorbed in the intestinal epithelial cell lines [2].

In this context, some strategies for the biotransformation of soy products in association or not with the use of probiotics or fermented food supplements have been investigated in order to increase the active compound bioavailability of interest. Dietary approaches can modify the microbiota profile while the microbiota profile can improve the bioavailability of aglycone or its secondary metabolites [3]. Ninety-two strains of lactic acid bacteria and bifidobacteria, like strains of the *Lactobacillus*, *Enterococcus*, *Lactococcus*, and *Bifidobacterium* genera, were analyzed for their ability to produce aglycone forms from isoflavones being able to produce daidzein, genistein, and glycitein [4].

In addition to fermentative bioprocesses, the use of enzymes, such as β-glucosidase, has also been shown to be efficient in the hydrolysis of glycosides that are poorly absorbed by the intestine. The β-glucosidases isolated from microbial sources (*Aspergillus niger*) and lima beans (*Phaseolus lunatus*) resulted in satisfactory conversion rates of daidzin and genistin in their respective aglycone forms [5].

In the present study, we used the commercial β-glucosidase enzyme to biotransform a soymilk extract in association or not with a commercial probiotic mix containing *Lactobacillus acidophilus*, *L. casei*, *Lactococcus lactis*, *Bifidobacterium bifidum*, and *B. lactis* strains followed by in vitro gastrointestinal digestion. In a previous prospective study, the same probiotic mix was used associated with a supplement rich in isoflavones within menopausal women, being able to promote the growth of bacteria associated with the metabolism of isoflavones and considered equal producers, such as *Faecalibacterium prausnitzii*, *Roseburia faecis*, *Alkaliphilus crotonatoxidans*, and *Blautia coccoide* [6]. Resuming, the main objective was to determine the effects of biotransformation processing on the bioaccessibility of soy isoflavones, evaluating the phenolic content and profile antioxidant capacity of different soymilk extracts obtained.

## 2. Materials and Methods

### 2.1. Materials

The soybeans (*Glycine max*; IAC Foscarin 31 variety) were obtained at the Agronomic Institute of Campinas, Campinas, SP, Brazil. A probiotic mix (Simfort^®^ Vitafor LOT: 10515) containing *Lactobacillus acidophilus*, *Lactobacillus casei*, *Lactococcus lactis*, *Bifidobacterium bifidum*, and *Bifidobacterium lactis* (10^9^ CFU, in 2 g) was used. The isoflavone standards (daidzin, genistin, glycitin, daidzein, genistein, and glycitein), β-glucosidase, pepsin from porcine gastric mucosa, pancreatin from a porcine pancreas, bile salts, gallic acid, tannic acid, sodium alginate, 2,2′-azobis(2-methylpropionamidine) dihydrochloride (AAPH), ρ-nitrophenyl-β-D-glucopyranoside (ρ-NPG), Trolox^®^, and fluorescein were purchased from Sigma-Aldrich (Steinheim, Germany). Folin–Ciocalteu’s reagent was purchased from Dinamica (Sao Paulo, Brazil). Methanol (99.9%, HPLC 10020 Appl Microbiol Biotechnol (2020) 104:10019–10031 grade) was obtained from J.T. Baker^®^ (Center Valley, PA, USA). All other chemicals were obtained through commercial suppliers.

### 2.2. Methods

#### 2.2.1. Processing of Soymilk Extracts

##### Preparation of Soymilk Extract

Soymilk extracts were produced from soybeans according to previous studies [7,8]. The lyophilized soymilk extract was suspended in sterile water (200 mg/mL) and used as the soymilk control (CT).

##### Enzymatic Biotransformation Process

The soymilk extract, prepared and lyophilized, was reconstituted in autoclaved water (200 mg/mL). The commercial β-glucosidase, Naringenase from Sigma/N1385-2GG, presenting the specific enzymatic activity of 10 U/g of the enzyme (3.6 mg/mL of the extract), was incubated for 10 min at 50 °C, pH 6.5. After incubation, the samples were shaken for 45 min at 50 °C. Then, the reaction was stopped in an ice bath for 15 min. Finally, the samples (3.6 mg/mL, using 54 mg for 15 mL of the final volume) were frozen for subsequent lyophilization. These samples were called enzymatically treated soymilk (ET).

##### Fermented Biotransformation Process

The commercial probiotic mix (Simfort^®^ Vitafor, São Paulo, Brazil, LOT: 10515) containing *Lactobacillus acidophilus*, *L. casei*, *Lactococcus lactis*, *Bifidobacterium bifidum*, and *B. lactis* was added to MRS-Broth–0.05%-cysteine (250 mg Simfort in 25 mL of the culture medium) and incubated at 37 °C overnight in an anaerobic flask. An aliquot from the first incubation was added to a new inoculum (1st passage) and new passages were performed until each sample achieved 10^9^ CFU. After activating the mix, 1% (*v*/*v*) of the culture inoculum was added in the soymilk extract reconstituted and incubated at 37 °C and 50 RPM for 24 h in an anaerobic flask.

After incubation, samples were frozen and lyophilized. These samples were called fermented soymilk (F).

##### Enzymatic Treatment Followed by Fermented Bioprocess

Samples obtained after enzymatic treatment (ET) were submitted to an additional step of probiotic fermentation as described in the “Fermented Biotransformation Process” section. These samples were called enzymatically treated and fermented soymilk (ET + T).

All reactions were performed in triplicate. As the control (CT), soymilk was submitted to the same conditions of both enzymatic and fermented biotransformation without enzyme and microorganism addition.

#### 2.2.2. *In Vitro* Digestion Assay

Samples from each experimental condition (CT, ET, F, and ET + T) were subjected to an in vitro digestion assay according to Minekus et al. [9]. For that, samples were reconstituted in autoclaved water (200 mg/mL) and submitted to a simulation of digestion of the oral, stomach, and small intestine stages, using their respective digestive fluids, as well as pH adjustment, temperature, and time in each experimental condition, as shown in Figure 1. Subsequently, the samples were lyophilized and frozen for later use.

#### 2.2.3. Analysis of Isoflavone Metabolites

Aliquots of samples (CT, ET, F, and ET + T) obtained before and after in vitro digestion were centrifuged for 10 min at 13,000× *g* and the fractions obtained were stored at 20 °C for quantification of isoflavone metabolites. For this, the aliquots were mixed with 5 mL of methanol (80%) at 25 °C for 2 min at 500 rpm. The mixture was centrifuged at 5000 rpm for 15 min and the supernatant was filtered through a 0.45 μm membrane before being analyzed with High Performance Liquid Chromatography (HPLC). Analyses were performed on a Dionex UltiMate 3000 Liquid Chromatograph (Thermo Fisher Scentific, Walham, MA, USA), equipped with an RP18 XTerra^®^ column (Waters, 5 μm, 4.6 × 150 mm) maintained at 30 °C. Detection was performed using a UV/VIS detector (DAD-3000) and the mobile phases A (water/formic acid, 99.9: 0.1 *v*/*v*) and B (methanol). Gradient elution was 20% B (0–15 min), 20–80% B (15–75 min), 80–100% B (75–80 min), 100–20% B (80–90 min), and 20% B (90−95 min), with a flow rate of 0.5 mL min^−1^. Spectra were recorded at 190 and 480 nm, and chromatograms were processed at 254 nm. Isoflavonoids were identified by comparing their retention time and absorbance spectrum. The quantification of individual bioactive compounds was performed using standard calibration curves (daidzin, genistin, glycitin, daidzein, genistein, and glycitein). Results were expressed as μg of each isoflavonoid compound/mL of the sample.

#### 2.2.4. Analysis of Total Phenolics

The total phenolic content (TPC) of the extracts was measured according to the Folin–Ciocalteu spectrophotometric method as described previously [8]. For that, CT, ET, F, and ET + T samples were analyzed in triplicate and the results were calculated using a calibration curve constructed using the gallic acid standard (between 300 μg/mL and 16 μg/mL). The reaction was determined at 725 nm using a spectrophotometer (DU 640, Beckman Coulter, Pasadena, CA, USA). The results were expressed as mg of gallic acid equivalents (GAE)/mL of the reconstituted sample.

#### 2.2.5. Analysis of Antioxidant Capacity

The analysis of antioxidant capacity was determined with the ORAC and FRAP assays, as previously described [8]. For this purpose, Trolox was used as a standard in the construction of the calibration curve at a concentration of 1500–1.5 μmol/mL for ORAC and 1500–15 μmol/mL for FRAP, and the results were expressed in μmol Trolox equivalents (TE)/mL of the reconstituted sample triplicate (CT, ET, F, and ET + T). ORAC assays were performed on a Fluostar Optima Microplate reader (FLUOstar OPTIMA; BMG Labtech, Ortenberg, Germany), which was equipped with fluorescence filters at 37 °C for 75 min with a 485 nm excitation filter and 520 nm emission filter. The results were calculated as the difference between the area under the fluorescein decay curve of the samples and the blank (net AUC), using regression equations between the net AUC and the concentration of the samples.

For the FRAP assay, the fluorimeter was set to 37 °C and the absorbance of the mixture was measured at 595 nm in a kinetic reading for 10 min (7 cycles of 88 s).

#### 2.2.6. Statistical Analysis

Results were expressed as the mean ± standard error of the mean (SEM) in triplicate. Statistically significant differences were determined using an analysis of variance (ANOVA), followed by Tukey’s post hoc test for multiple comparisons using GraphPad Instat 3.1 (GraphPad Software, Inc., La Jolla, CA, USA). *p* < 0.1 was considered significant for all analyses. The statistical analysis and graphics were performed using GraphPad Prism version 6.0 (GraphPad Software, USA).

## 3. Results and Discussion

### 3.1. Profiling of Bioactive Isoflavones

Different biotransformation processes with enzymes and/or probiotic strains have been used as a strategy to convert the glycosylated form of isoflavones into their respective aglycones [10,11].

In this study, the isoflavones’ profile in samples after the enzymatic treatment, fermented bioprocess, or enzymatic-followed fermented bioprocess was evaluated with HPLC-DAD, as shown in Figure 2, and they are illustrated in the chromatograms in Figure 3.

Daidzin, a glycosylated form, was detected in a higher concentration only in control and enzymatically treated soymilk (30.78 µg/mL of the CT and 27.30 µg/mL of ET; Figure 2A and Figure 3A,C). Daidzin levels were undetected in the fermented soymilk and were detected in a minor concentration in enzymatically treated followed by fermented soymilk if compared to the soymilk control (from 30.78 µg/mL of the CT to 4.91 µg/mL of ET + T; Figure 2A and Figure 3A,G).

The conversion of daidzin to daidzein, its respective aglycone, was not observed in enzymatically treated soymilk (2.42 µg/mL of the CT and 9.8 µg/mL of ET; Figure 2B and Figure 3A,C). However, in fermented and enzymatically treated followed by fermented soymilk, an increase in daidzein was detected (from 2.42 µg/mL of CT to 26.41 µg/mL of F and 44.82 µg/mL of ET + T; Figure 2B and Figure 3A,E,G).

Genistin, a glycosylated form, was detected in different concentrations after each bioprocess (from 28.86 µg/mL of CT to 43.54 µg/mL of ET; 11.50 µg/mL of F; Figure 2C and Figure 3A,C,E), being undetected in enzymatically followed by fermented treated soymilk. Genistein, the aglycone form of genistin, was also detected in different concentrations after each bioprocess, suggesting that the conversion of genistin into genistein occurred in all bioprocesses (from 2.03 µg/mL of CT to 9.52 µg/mL of ET; 20.94 µg/mL of F; 37.23 µg/mL of ET + T; Figure 2C and Figure 3A,C,E,G).

Glycitin, another glycosylated form, was detected in a lower concentration in fermented and enzymatically treated followed by fermented soymilk if compared to the soymilk control (from 6.30 µg/mL of CT to 9.61 µg/mL of ET; 3.32 µg/mL of F and 2.78 µg/mL of ET + T; Figure 2E and Figure 3A,C,F,G). However, regarding glycitein, its respective aglycone, the concentrations had no significant changes between treatments (3.37 µg/mL of CT to 3.44 µg/mL of ET; 3.52 µg/mL of F and 3.26 µg/mL of ET + T; Figure 2F and Figure 3A,C,F,G.

In previous studies by our research group, the β-glucosidase enzyme was also used to ferment soymilk, resulting in the conversion of glycosidic isoflavones into aglycones, and the enzymatic process was previously optimized [10,11].

Fermentation treatment using combinations of different probiotic strains results in variations in the rate of conversion of glycosylated isoflavones to aglycones. A previous study that used the combination of *Bifidobacterium lactis* and *Lactobacillus casei* resulted in an increase in the daidzein content (from 1 to 7/100 g of sample dry weight) and genistein (from 2 to 5/100 g of the dry weight sample) [8]. In another study, a combination of the same strains of *B. lactis* (BLC1) and *L. casei* (MB151) resulted in an increase in daidzein, genistein, and glycitein around 10%, 225%, and 36%, respectively, compared to the control [12]. A previous study of our research group was carried out using different bacterial strains to ferment a soymilk extract, including *Streptococcus thermophilus* (YF-L811), *Lactobacillus delbrueckii* ssp. *bulgaricus* (LB-340), *Bifidobacterium* ssp. *lactis* (Bb-12), and *Lactobacillus acidophilus* (LA-05), that resulted in an increase 53 times greater than that of an unfermented extract [7]. In another study, seven strains of lactic acid bacteria and bifidobacteria, including *Bifidobacterium animalis* ssp. *lactis* BLC 1, *Lactobacillus acidophilus* LA 3, *Streptococcus thermophilus* ST 066, *Lactobacillus casei* MB151 (ATCC 334), *Lactobacillus delbrueckii* ssp. *bulgaricus* MB153 (ATCC 9649), *Lactobacillus rhamnosus* MB154 (ATCC 7469), and *Lactobacillus kefiri* CBMAI212, were used. The fermentation process using all strains was able to reduce the content of isoflavone glycosides with a consequent increase in aglycones, varying between 6 and 8 times for daidzein, 5.6 and 7.4 times for genistein, and 2.3 and 3.8 times for glycitein, in comparison with the control [13]. Other studies with soy-based foods used some bacterial cultures, such as *Lactobacillus delbrueckii* ssp. *Bulgaricus*, *L. acidophilus*, *L. plantarum*, *L. casei*, *L. fermentum*, and *S. thermophilus*, and several species of *Bifidobacterium* were also responsible for increasing the content of aglycone isoflavones [14,15,16].

Despite some differences in the rate of isoflavone conversion observed in these studies, all strains used were able to efficiently hydrolyze glycosides into their aglycone forms. In this way, we can associate these variations due to the activity of β-glucosidase inherent in the microorganisms used to carry out the fermentation process, since β-glucosidase is an enzyme involved in the hydrolysis of the β-glycosidic bond present in isoflavone glycosides, releasing the bioactive aglycone forms [17]. In any case, all microorganisms used were efficient in the ability to deglycosylate soy isoflavones, as also demonstrated by other authors [16,18,19]. In addition to properly selecting microorganisms to be used as biotransformers in soy products, it is necessary to consider the physicochemical parameters (i.e., pH, acidity, and viscosity) to obtain adequate cell growth of the chosen strains [20].

However, in this study, the combination of enzymatically treated and fermented soymilk resulted in an 18-fold increase for dadzein and genistein compared to the soymilk control, indicating a synergistic effect between the enzymatic and fermentative bioprocess, whereas the fermented treatment alone resulted in an 11- and 10-fold increase for dadzein and genistein, respectively.

Overall, in some studies mentioned above, the combination of fermentative and enzymatic bioprocesses using the enzyme tannase or β-glucosidase resulted in a superior synergistic effect compared to bioprocesses alone [12,13].

It is known that the stability and release of isoflavones can be altered with the gastrointestinal digestion process due to the acidic pH of the stomach and the chemical composition of the digestive fluids [21]. In this context, we investigated the bioaccessibility of compounds present in a soymilk extract and their treatments by simulating in vitro gastrointestinal digestion at oral, gastric, and intestinal levels.

After in vitro digestion, the daidzin content was reduced in all soymilk extracts compared to the same sample before digestion (Figure 2A), but a higher daidzein content was detected for fermented treatment soymilk (of 5.13 µg/mL of CT; of 15.0 µg/mL for F), that is, three times greater (Figure 2B).

The genistin and genistein content was reduced in all samples compared to the same sample before digestion (Figure 2C and Figure 2D, respectively). However, fermented and enzymatically/fermented soymilk treatment presented the higher genistein content if compared with the soymilk control after in vitro digestion (from 3.82 µg/mL of CT to 13.74 µg/mL of F and 9.09 µg/mL of ET + T; three times and two times greater than CT, respectively; Figure 2D and Figure 3B,F,H).

Glycitin, another glycosylated form, was detected in a lower concentration in fermented and enzymatically treated followed by fermented soymilk if compared to the soymilk control (from 7.32 µg/mL of CT to 7.56 µg/mL of ET; 3.32 µg/mL of F and 2.78 µg/mL of ET + T; Figure 2E and Figure 3B,D,F,H). However, regarding glycitein, its respective aglycone, the concentrations had no significant changes between treatments (3.37 µg/mL of CT to 3.44 µg/mL of ET; 3.52 µg/mL of F and 3.26 µg/mL of ET + T; Figure 2F and Figure 3B,D,F,H).

Glycitin content was not detected in soybean extracts that underwent enzymatic, fermentative, and enzymatic bioprocesses followed by fermentative processes (Figure 2E). Furthermore, glycitein, its respective aglycone, was detected only in control and fermented soymilk, but no significant difference was observed (Figure 2F and Figure 3B,D,F,H).

The presence of aglycones after the in vitro digestion simulation indicates that these isoflavones would be available for absorption. In a previous study by Queirós et al. [22], the enzymatic extraction of soy flour extracts using different enzymes (protease, tannase, and cellulase) or their combination resulted in a significant increase in aglycones after in vitro gastrointestinal digestion, except that produced with the use of protease alone. Another study using kefir-fermented soymilk also resulted in increased aglycone isoflavone content after in vitro gastrointestinal digestion [23].

Considering β-glucosidase is an enzyme involved in the hydrolysis of the β-glycosidic bond present in the glycosidic isoflavone, releasing the bioactive forms of the aglycones [17,24], and as mentioned before, although enzymatically treated soymilk with commercial β-glucosidase was not able to efficiently convert the glycosidic forms into aglycones, the results obtained in this study may be associated with the β-glucosidase activity of the probiotic microorganisms used to carry out the soymilk fermentation process.

### 3.2. Phenolic Compounds and Antioxidant Activity of Soymilk Extracts

Soy isoflavones are considered an important class of antioxidant compounds, with the aglycone forms and their secondary metabolites having greater antioxidant potential compared to other isoflavonoids [25]. The role of antioxidant compounds present in food or in the body is to delay, control, and prevent oxidative processes that lead to food deterioration and the spread of degenerative diseases in the body. Therefore, measuring the antioxidant capacity of foods is essential not only to ensure the quality of functional foods but mainly to study the efficiency of food antioxidants in the prevention and treatment of diseases related to oxidative stress [26].

In this study, the total phenolic content using the Folin–Ciocalteu test, as well as the antioxidant capacity using the ORAC and FRAP assays, were evaluated in the enzymatic bioprocess with β-glucosidase and/or fermentation with probiotics (CT, ET, F, and ET + T), as shown in Figure 4A–C. The results presented indicated that ET soymilk followed by F had a higher content of total phenolics before in vitro digestion (from 3.48 mg/mL of CT to 4.48 mg/mL of ET + T; Figure 4A). The samples that were submitted to the in vitro digestion assay resulted in the maintenance of the total phenolic content for all bioprocesses when compared to CT (from 2.81 mg/mL for the CT to 3.80 mg/mL for ET; 3.84 mg/mL for F and 4,03 mg/mL of ET + T; Figure 4A). Moreover, we observed a reduction in total phenolic content in CT after the digestion process, which resulted in the highest significant values (*p* < 0.05) for all soymilk extracts before and after digestion, as shown in Figure 4A.

The ET + T process also resulted in the highest antioxidant capacity represented by the ORAC assay (from 120 mg/mL of CT to 151 mg/mL of ET + T; Figure 4B) and by the FRAP assay (from 285 µL/mL of CT to 317 µL/mL of ET + T; Figure 4C) compared to CT before in vitro digestion. However, we did not observe significant changes between groups of treatments after the in vitro digestion assay (Figure 4B,C).

On the other hand, in the study by Queirós et al. [22], the concentration of total phenolic compounds and the antioxidant capacity were higher after in vitro gastrointestinal digestion compared to undigested samples. Such differences can be explained by the different specificities of enzyme action, since different enzymes were used between the current studies and the study mentioned above, as well as variations in processing conditions, which may change the rate of absorption and metabolism of isoflavones and consequently, in their biological activities [27]. Furthermore, the methods adopted in this study measure the different compounds and, therefore, cannot be compared with each other.

Additionally, the activation of the probiotic bacteria used in this study proved to be efficient not only in converting the glycosylated forms of isoflavones into aglycones, but also in improving the bioaccessibility of these aglycones and antioxidant capacity. Our data corroborate previous studies, in which fermented soy products showed greater antioxidant capacity than non-fermented ones [28,29].

Previous studies by our research group mentioned above demonstrated that the enzymatic and/or fermentative bioprocesses used associated the conversion of glycosidic isoflavones into aglycones with a higher content of total phenolics and the antioxidant capacity of a soymilk extract [7,13] and soy flour extract [22]. In the first study, all bioprocesses tested resulted in a significant increase in total phenolic content and antioxidant capacity using the ORAC and DPPH assays, with the best results being observed in soymilk that was subjected to biotransformation with the tannase enzyme [7]. In the second study, the combination of microbial fermentation and tannase biotransformation also resulted in increased total phenolic content and antioxidant activity using the ORAC and FRAP assays [13]. The antioxidant capacity of soymilk using Lactobacillus fermentation, of enzymatic biotransformation with tannase and enzymatic biotransformation followed by fermentation, was also evaluated previously, resulting in increased antioxidant capacity using ORAC and FRAP assays for all bioprocesses, with superior results being observed for enzymatic biotransformation followed by fermentation [8]. Finally, in the soybean flour extract, all the bioprocesses used resulted in an increase in the release of total phenolics and in the antioxidant capacity, measured with the scavenging of the ABTS+ radical, with the exception of the cellulase enzyme in isolation [22].

## 4. Conclusions

The proposed biotransformation processing using enzymatic and probiotic fermentation alone and combined was effective to convert glycosidic isoflavones into aglycone forms. All the processing performed resulted in the richest phenolic extracts and higher antioxidant capacity before the in vitro digestion. This work was important to demonstrate that during in vitro digestion processing, the antioxidant capacity can be modified, as well as the phenolic profile, in extracts. The bioprocessing can be a good approach to increase aglycones and antioxidant capacity in extracts, but the extract must be prepared and bioactives protected by encapsulation or emulsified before digestion processing. This is a real challenge and must be investigated for future works.

Therefore, the biotransformation process before in vitro digestion was relevant for maintaining the bioactive potential of the extracts, by increasing the antioxidant capacity and the profile of bioaccessible isoflavones before digestion.

So, the combined isoflavone biotransformation proposed in this work has the potential of enriching the functional and nutritional properties of fermented soy-based products, increasing the positive effects for consumers for future applications.

## Figures and Tables

**Figure 1 foods-12-03401-f001:**
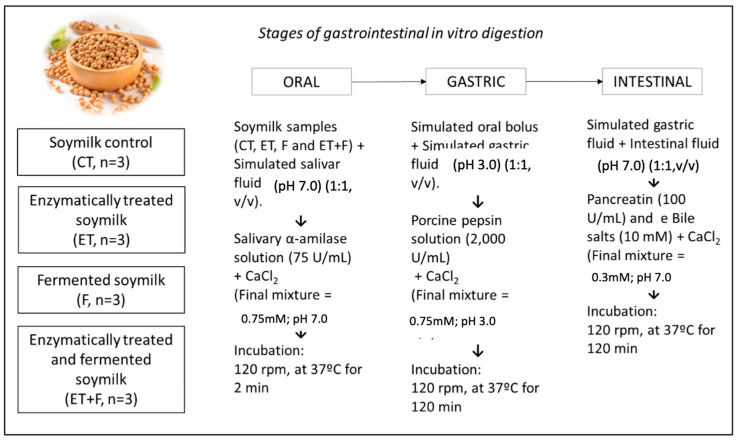
Methodological scheme of stages of gastrointestinal in vitro digestion of soymilk extracts during oral, gastric, and intestinal phases.

**Figure 2 foods-12-03401-f002:**
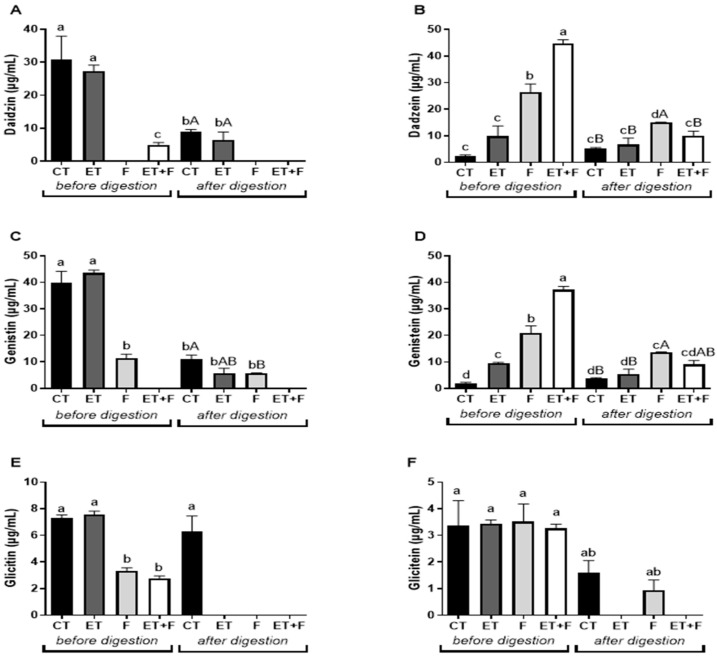
Concentration of soymilk isoflavones before and after gastrointestinal *in vitro* digestion with LC-MS/MS. (**A**): Daidzin concentration before and after soybean extract *in vitro* digestion. (**B**): Dadzein concentration before and after soybean extract *in vitro* digestion; (**C**): Genistin concentration before and after soybean extract *in vitro* digestion; (**D**): Genistein concentration before and after soybean extract *in vitro* digestion; (**E**): Glicitin concentration before and after soybean extract *in vitro* digestion and (**F**): Glicitein concentration before and after soybean extract *in vitro* digestion. Abbreviations: CT, soymilk control; ET, enzymatically treated soymilk; F, fermented soymilk; ET + T, enzymatically treated and fermented soymilk. Values are presented as µg of sample/mL of extract and mean ± standard deviation (*n* = 3). ^a,b,c,d^ Different lowercase letters indicate a significant difference between bioprocess methods before and after *in vitro* digestion. ^A,B^ Different capital letters indicate a significant difference between bioprocess methods after *in vitro* digestion, as indicated with the Tukey’s test (*p* < 0.1).

**Figure 3 foods-12-03401-f003:**
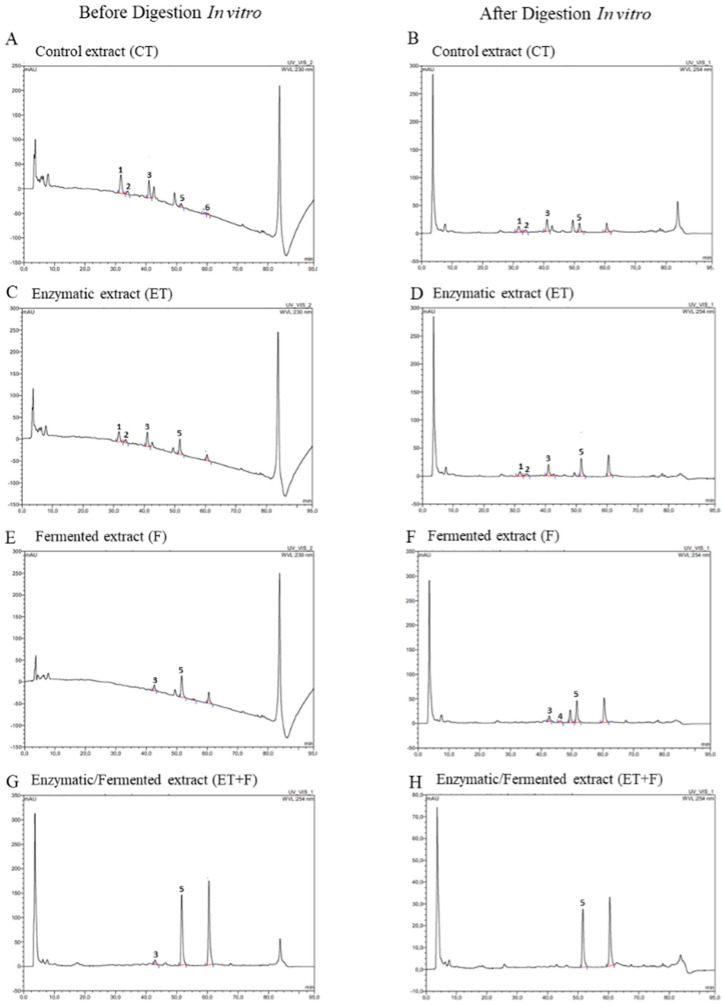
Representative chromatograms obtained with HPLC-DAD before and after gastrointestinal in vitro digestion with LC-MS/MS from soymilk control (CT; (**A**,**B**)); enzymatically treated soymilk (ET; (**C**,**D**)); fermented soymilk (F; (**E**,**F**)); and enzymatically treated and fermented soymilk (ET + T; (**G**,**H**)). The numbers represent the respective isoflavones: daidzin (1), glycitin (2), genistin (3), glycitein (4), daidzein (5), and genistein (6).

**Figure 4 foods-12-03401-f004:**
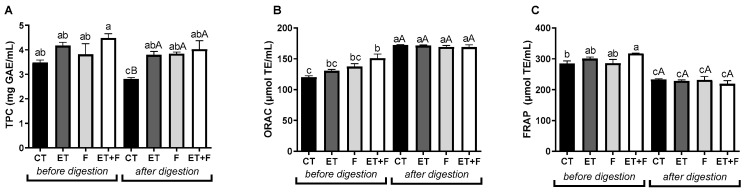
Total phenolic content (**A**) and antioxidant capacity (**B**,**C**) before and after gastrointestinal *in vitro* digestion. Abbreviations: TPC, total phenolic content; GAE, gallic acid; TE, Trolox; CT, soymilk control; ET, enzymatically treated soymilk; F, fermented soymilk; ET + T, enzymatically treated and fermented soymilk. Values are presented as mg of sample/mL of extract (**A**) and µmol TE/mL (**B**,**C**) and mean ± standard deviation (*n* = 3). ^a,b,c^ Different lowercase letters indicate a significant difference between bioprocess methods before and after *in vitro* digestion. ^A^ Different capital letters indicate a significant difference between bioprocess methods after in vitro digestion, as indicated with the Tukey’s test (*p* < 0.1).

## Data Availability

The data used to support the findings of this study can be made available by the corresponding author upon request.

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
