# Peer review of "Bioaccessibility Evaluation of Soymilk Isoflavones with Biotransformation Processing"

_foods, 2023, doi:10.3390/foods12183401_

Round 1

Reviewer 1 Report

In this manuscript, the authors present the processed soymilk through commercial beta-glucosidase treatment, probiotic fermentation, and combined enzymatic treatment with fermentation. The contents of isoflavones (in glycosylated form or aglycone), as well as the total phenolic content, antioxidant capacity, were evaluated before and after in vitro simulated gastrointestinal digestion.

Some questions:

1. The “Chromatograms obtained by HPLC-DAD…” should be put forward in Fig.2 to make readers easier to read. And since each data were collected in triplicate, the title of figure legend should be “Representative chromatograms obtained ….”

2. From the above chromatogram data, the concentration of each isoflavones was determined (as described in material and methods). But the data did not match with Figure 2 “Concentration of soymilk isoflavones ….”. Eg. The concentration of geinstin (compound labeled (3) in Figure 3) in ET is clearly lower than CT, but it is not in Figure 2C.  In Figure 3, even the peak of glycitin cannot be clearly labeled in chromatogram F and ET+F, and how can the concentration of these compounds be determined to be about 2.5 ug/ml in Figure 2?  And so forth.

3. What kind of beta-glucosidase be used in this study? What’s its enzymatic specific activity? Since it is batch specific, authors should provide this information. In line 97, what ratio of enzyme used in enzymatic treatment? Also in line 98, what samples were processed ? (After incubation, the samples were shaker ….)  The volume of samples? 

4. The methods about photometric detection in section 2.2.4, 2.2.5 were not clearly described. Eg, Line 144, Spectra were recorded from 190nm to 480nm,…  Line 154-155, The reaction was determined for absorbance at 725nm using a photometer… (Du 640 is a spectrophotometer, not a fluorometer)  Also in line 168, the condition of the photometer was set to 37°C and the ….  Line 167, please provide the correct fluorometer used (machine type, brand…)

5. The commercial Simfor probiotic mix containing several types of lactic bacteria. Its viable count is dependent on each batch and storage process. As mentioned in the material and methods section 2.2.1, line 103-111, two amplification passages were performed before adding to reconstituted soymilk extract. How is the (quality) control of the inoculum? Was its OD measured? What size of the fermentation volume performed?

6. In Figure 2 and Figure 4, the description of statistical significances are ambiguous. Is “a” means the comparison of data in CT between before and after in vitro digestion?  Is “b” means the comparison between CT and ET? 

7. From the data in Figure 2, after in vitro digestion, the concentration of 3 isoflavone aglycones in ET, F, ET+F were all higher than CT. However, in Figure 4, after in vitro digestion, only TPC shows similar results with increasing aglycone conc. ORAC and FRAP did not show similar trend. It seems not enough to support that combined enzymatic and biotransformation for “improved” antioxidant capacity.  What is the authors discussion? 

8. The genus and species of the bacteria should be in italics.

9. Each measured quantity should have their corresponding unit. It is not suitable to omit it. Eg. Line 22, 3.48 mg/ml for ….; line 23, 120 mg GAE/ml for control and …., and so forth.

10. Usage of enzyme treatment alone (ET), fermentation by Simfort alnoe (F), combined enzymatic process and fermentation (ET+F), for the description in main context probably easier for reader to read.

Author Response

Attach

Reviewer 2 Report

Plagiarism must be reduced its 35%

Comment 1: The title should be properly rephrased.

Comment 2: Use the complete scientific name when used for the first time, then abbreviate it. Don’t use authority in abbreviated scientific names.

Comment 3: Use the proper format for the abstract, i.e., a brief intro, methodology, significant results, conclusions, along with recommendations, and finally its applications. Don’t use headings in the abstract. Delete irrelevant details. Abstracts should NOT exceed 250 words.

Comment 4: Don’t repeat the keywords used in the title. Use unique keywords, which increase the worth of your article on search engines.

Comment 5: Add proper information in the intro. Rewrite it again by adding the following: i.e., brief intro, problem statement, significance at the national and international level, research gap, objectives, etc. Add the latest references in the intro relevant to your work. Follow Journal format throughout the article for references.

Comment 6: Thoroughly revise the methodology; it's ambiguous, lacks proper references, and can’t be replicated.

Comment 7: Rewrite it (the statistical analysis) again properly by adding references.

Comment 8: The results section was not in accordance with the figures and tables; there was a lot of difference; rewrite it again; it is so confusing. Use the same heading format in discussion as well, as discussion was totally lacking in the article, and without comparison with the available date, it has no worth.

Comment 9: The conclusion section was not in accordance with the proposed objectives. Recommendation and future application are also missing it.

Comment 10: Follow journal formatting for references; there are a lot of mistakes here.

Reduce the number of references to 40. Delete irrelevant and older ones; use only the latest.

Comment 11: Use scientific names throughout the article.

Comment 12: Add proper standard error bars for data comparisons.

Moderate editing of English language required

Author Response

Attach

Round 2

Reviewer 1 Report

Comments

Line 101, since the Simfort is a mix of several kinds of lactobacillus species, the each species possess its own growing rate, thus the population of these bacteria may not be the same at different culture time. In this manuscript, the control of this measurements, such as culturing time or the OD of the culture broth, is totally missing. This makes it very difficult for other group to repeat the experiment. The control of this measurements is not a scientific performance as “was observed with the medium becoming cloudy “, the OD or culture time must be provided.

As described in the Methods section 2.2.3, the each isoflavone was quantified by HPLC spectra. As shown in In Figure 2C, the contents of genistin before digestion was measured as about 29ug/ml in CT group, this value was obviously lower than that of ET group with about 42ug/ml. However, as shown in HPLC spectra Figure 3A (CT) and 3C (ET), the area of peak labeled 3 (as described is genistin) in CT group (Fig 3A) was shown to be a little larger than that in ET group (Fig 3C). The data is not consistent

Similar situation was happened as glycitin that was labeled 2 in HPLC spectra in “before digestion” groups.

Reviewer 2 Report

Accept in present form
